# Accurate point-of-care serology tests for COVID-19

**Charles F. Schuler, IV**[1,2]*, **Carmen Gherasim**[3], **Kelly O'Shea**[1,2], **David M. Manthei**[3], **Jesse Chen**[1,4], **Don Giacherio**[3], **Jonathan P. Troost**[5], **James L. Baldwin**[1,2☯], **James R. Baker, Jr**[1,2,4☯]

**1** Division of Allergy and Clinical Immunology, Department of Internal Medicine, University of Michigan, Ann Arbor, MI, United States of America, **2** Mary H. Weiser Food Allergy Center, University of Michigan, Ann Arbor, MI, United States of America, **3** Department of Pathology, University of Michigan, Ann Arbor, MI, United States of America, **4** Michigan Nanotechnology Institute for Medicine and Biological Sciences, University of Michigan, Ann Arbor, MI, United States of America, **5** Michigan Institute for Clinical and Health Research, University of Michigan, Ann Arbor, MI, United States of America

☯ These authors contributed equally to this work.
* schulerc@med.umich.edu

## Abstract

### Background

As COVID-19 vaccines become available, screening individuals for prior COVID-19 infection and vaccine response in point-of-care (POC) settings has renewed interest. We prospectively screened at-risk individuals for SARS-CoV-2 spike and nucleocapsid protein antibodies in a POC setting to determine if it was a feasible method to identify antibody from prior infection.

### Methods

Three EUA-approved lateral flow antibody assays were performed on POC finger-stick blood and compared with serum and a CLIA nucleocapsid antibody immunoassay. Variables including antibody class, time since PCR, and the assay antigen used were evaluated.

### Results

512 subjects enrolled, of which 104 had a COVID-19 history and positive PCR. Only three PCR-positive subjects required hospitalization, with one requiring mechanical ventilation. The POC results correlated well with the immunoassay (93–97% sensitivity) and using serum did not improve the sensitivity or specificity.

### Conclusions

Finger-stick, POC COVID-19 antibody testing was highly effective in identifying antibody resulting from prior infections in mildly symptomatic subjects. Using high-complexity serum immunoassays did not improve the screening outcome. Almost all individuals with COVID-19 infection produced detectable antibodies to the virus. POC antibody testing is useful as a screen for prior COVID-19 infection, and should be useful in assessing vaccine response.

**Data Availability Statement:** The study includes potentially identifying and sensitive patient information, governed under HIPAA, therefore access is limited to by request by our Institutional Review Board and Office of Research. The

University of Michigan Data Access Office can be reached for information on data requests by phone at 734-615-2100 or email at DataOffice@umich.edu.

**Funding:** The funders had no role in study design, data collection and interpretation, manuscript drafting, manuscript editing, or the decision to submit the work for publication. This work was supported by the University of Michigan (Institutional Funding, COVID-19 Innovation Grant), the National Institutes of Health (UL1TR002240), and through related sponsored projects from Healgen Scientific (healgen.com) and Access Bio Inc (accessbiodiagnostics.net). In addition, lateral flow assays were provided at no cost by Healgen Scientific, Access Bio Inc, and Autobio Diagnostics Co Ltd (https://www.autobio.com.cn/en/).

**Competing interests:** Dr. Schuler reported salary and other support from the Mary H. Weiser Food Allergy Center and the Taubman Innovation Institute at UM, as well as a UM COVID-19 Innovation Grant; he has also received sponsored project support from Healgen Scientific and Access Bio Inc. Dr. Troost was supported in part by the National Center for Advancing Translational Sciences (NCATS) for the Michigan Institute for Clinical and Health Research (UL1TR002240). Dr. Baldwin and Dr. Baker reported salary support from a UM COVID-19 Innovation Grant and sponsored project support from Healgen Scientific and Access Bio Inc. Dr. Gherasim, Dr. O'Shea, Dr. Manthei, Mr. Chen, and Dr. Giacherio reported no Competing Interests. This does not alter our adherence to PLOS ONE policies on sharing data and materials.

**Abbreviations:** COVID-19, Coronavirus disease 2019; SARS-CoV-2, severe acute respiratory syndrome coronavirus-2; PCR, polymerase-chain-reaction; RT-PCR or "PCR", reverse transcriptase polymerase chain reaction; LFA, lateral flow assay(s); POC, point-of-care; U-M, University of Michigan; CareStart COVID-19 IgM/IgG or "CareStart", Access Bio; Anti-SARS-CoV-2 Rapid Test or "Autobio", Autobio Diagnostics; COVID-19 IgG/IgM Rapid Test Cassette or "Healgen", Healgen Scientific; EUA, Emergency Use Authorization.

# Introduction

Coronavirus disease 2019 (COVID-19), caused by severe acute respiratory syndrome coronavirus-2 (SARS-CoV-2), has caused a pandemic with millions of cases and deaths [1]. As effective vaccines become available, the question of screening individuals for prior COVID-19 infection has become relevant. Serological testing is the best accepted method to detect prior COVID-19 infection. In contrast to polymerase chain reaction testing, which is positive during acute infection, serologic testing identifies antibody responses to prior infection [2, 3]. Antibody assays have been useful for prevalence determinations; however, fuller validation of serologic testing to detect COVID-19 infection is still needed. Current assays use different SARS-CoV-2 detection antigens and have different testing conditions [2–4]. This is important since all current vaccines and those in development only involve spike protein. In addition, some assays identify both virus-specific IgM and IgG while others only identify IgG. The most efficient approach to screen for antibody is in point-of-care (POC) settings, but no assay is approved for this use.

CLIA laboratory-based antibody assays are expensive, require blood draws for serum, and need sample transport to a CLIA-approved facility. Rapid antibody tests using lateral flow assay(s) (LFA) solve these logistical problems but have had other issues. While initially loosely regulated their function has now been verified by a standardized EUA in the US requiring a prior FDA technical review. Despite this, the accuracy of these assays has continued to be questioned, and no test is currently approved for finger-stick POC use [5, 6]. Thus, real-world data are needed on LFA accuracy to identify prior COVID-19 in practical use; we therefore aimed to evaluate three COVID-19 LFA tests in a POC setting.

In this study, we prospectively evaluated the ability of LFA to detect antibody in a cohort at risk for COVID-19. We used LFA from three manufacturers who had obtained full EUA approval after evaluation by the FDA. We tested subjects for antibody in a POC setting and compared the POC LFA results with results from serum obtained at the same time that was analyzed separately. LFA results were also compared to a high-moderate CLIA laboratory immunoassay run on the same serum sample and compared to the subjects' clinical history and PCR results. The results suggest potential utility for LFA to screen for prior COVID-19 infection in the POC setting.

# Methods

## Study populations

This study was reviewed and approved by the U-M Institutional Review Board (IRB) and conducted according to the principles expressed in the Declaration of Helsinki. All participants provided informed written consent. Enrollment occurred between April 28, 2020 and October 2, 2020. All study activities occurred at the University of Michigan (U-M) Division of Allergy and Clinical Immunology, Department of Clinical Pathology, and Biomedical Science Research Building. All subjects were recruited via phone or email announcements with phone follow-up. Potential subjects were selected who had recently had a COVID-19 PCR test (see below) were contacted by phone. U-M healthcare workers were recruited via approved emails to work areas impacted by COVID-19 patient care.

Three populations were recruited.

1. Healthcare workers without defined COVID-19 status: Healthcare workers providing direct care for COVID-19 patients in any setting could enroll without a prior COVID-19 test.

2. Known COVID-19-Positive Group: Adults with demonstrated COVID-19 infection were eligible. COVID-19 infection was defined by reverse transcriptase polymerase chain

reaction (RT-PCR or "PCR") positivity for SARS-CoV-2 from a nasopharyngeal swab on a clinical sample. The EUA RT-PCR tests for COVID-19 were run at the U-M Clinical Pathology Laboratory and include testing by Abbot RealTime SARS-CoV-2 assay on m2000 systems and DiaSorin Simplexa COVID-19 Direct Kit on Liaison MDX. Subjects could enroll using a PCR test run outside U-M in a CLIA-certified laboratory if written verification was provided, including multiple tests performed at Viracor. Blood samples were taken at least 10 days from the onset of symptoms and positive PCR. Many of these subjects were healthcare workers.

3. Known COVID-19-Negative Group: Adults with a negative COVID-19 PCR within the prior 14 days and no known history of COVID-19 were eligible. Many were healthcare workers.

Exclusion criteria included immunodeficiency/immunosuppression with known primary or acquired immunodeficiency; anti-rejection therapy following solid organ or bone marrow transplant; biologic therapeutics such as tumor necrosis factor inhibitors; known malignancy and chemotherapy; systemic immunosuppressive therapy, including corticosteroids equivalent to 20 mg/day of prednisone for 2 weeks.

## Data collection

Data were collected in Research Electronic Data Capture software (REDCap, Vanderbilt University). Subjects reported demographics, past medical history, and medications. For known COVID-19-positive subjects, the infection course, travel history, known exposure to COVID-19, and, if applicable, the hospital course, including medications, laboratory and imaging, intensive care needs, mechanical ventilation, and complications, were reviewed. Any other illnesses that subjects had during the time-period after March 1st, 2020 were reviewed. The subjects' medical records were also reviewed to verify key data.

## Specimen collection and handling

Capillary blood samples by finger-stick were obtained with spring-loaded lancets using aseptic technique. Blood samples of 10 mL were collected via standard phlebotomy using aseptic technique into no-additive collection tubes and refrigerated at 4˚C until same-day processing. Samples were centrifuged at 2000 g for 10 minutes at 4˚C. Serum was transferred into 2 mL aliquots in glass screw-thread vials (DWK Life Sciences) and stored at -20˚C until analysis.

## Lateral flow assays

POC COVID-19 antibody tests were used from Access Bio (CareStart COVID-19 IgM/IgG or "CareStart"), Autobio Diagnostics Co Ltd (Anti-SARS-CoV-2 Rapid Test or "Autobio") and Healgen Scientific (COVID-19 IgG/IgM Rapid Test Cassette or "Healgen"). The CareStart detects both SARS-CoV-2 spike (receptor binding domain or RBD portion) and nucleocapsid antibodies, while the Autobio and Healgen detect SARS-CoV-2 spike (RBD) antibodies (Healgen detects S1 protein antibodies). Each test was run per each company's EUA—Instructions for Use [7, 8], except testing was done with both whole blood from finger-stick and serum from a blood draw and incubated as appropriate. Tests were considered positive if IgM and/or IgG were positive. A faint line was read as positive.

During the course of the study the Autobio had its EUA revoked related to poor performance of the IgM portion of the test; we stopped using this device at the request of our IRB [8, 9]. We also evaluated several other LFA that did not have FDA EUA approval. However,

preliminary analysis of these assays showed performance that was inadequate to include in the results.

### SARS-CoV-2 nucleocapsid antibody electrochemiluminescence immunoassay

Nucleocapsid antibody testing was performed on the Elecsys® (Roche) SARS-CoV-2 Total Antibody Assay ("immunoassay"), run on a Cobas e411 analyzer in the Clinical Core Laboratories at U-M, a CLIA-certified lab. The assay detects total SARS-CoV-2 nucleocapsid (N) antibodies via a sandwich electrochemiluminescence assay (ECLIA). Positive and negative controls were included with each run. A cutoff index (COI) of > 1 is used to report a positive result. The assay detects all SARS-CoV-2 N antibodies, including IgG [8]. This assay has been well-characterized by our institution previously [10].

### Statistical analysis

Descriptive statistics were provided using medians and interquartile ranges (for continuous variables) and frequencies and percentages (for categorical variables). Finger-stick assay results were compared to serum results, immunoassay, and PCR; serum results were also compared to the immunoassay and PCR. For finger-stick and serum, separate analyses were done by brand and antibody (i.e., IgM and/or IgG). 2x2 contingency tables frequencies and measures of diagnostic accuracy (i.e., sensitivity, specificity, positive and negative predictive value) were calculated for each comparison. Area under the curve (AUC) was calculated using logistic regression. Time between antibody test and PCR were compared between antibody positive and negative tests using an unpaired t-test. Two-sided $\alpha$ = 0.05 was used to assess statistical significance, and no adjustments were made for multiple comparisons as there were relatively few (by brand, by IgM vs. IgG, and by serum vs. POC) and were considered distinct comparisons. Analyses were based on complete-case analyses; participants with missing data for a given analysis were excluded. No outlier values for the variables included were observed (most data elements are "positive vs. negative"). Subgroup variation in agreement by age and occupation (HCW vs. general population) was tested using interaction terms in logistic regression models. 512 participants were enrolled. Assuming 15% would be COVID positive, and assuming a true AUC of 0.9, this sample would have 80% power to detect a confidence interval width of 0.09, with two-sided alpha = 0.05. Analyses were performed in SAS V9.4 (SAS Institute Inc., Cary, NC, USA) and Prism V8.0 (GraphPad Inc., San Diego, CA, USA).

## Results

We enrolled 512 subjects; 104 had PCR-confirmed COVID-19. 93 subjects had a known negative COVID-19 PCR within 14 days of enrollment. The remainder had no history of a PCR. No subjects had a known COVID-19 reinfection. The median age was 39 years old, 75% were female, and 88% were healthcare workers (Table 1). Among PCR-confirmed COVID-19 subjects, symptoms included fever >38˚C (53%), subjective fever (77%), chills (79%), myalgias (82%), headache (75%), diarrhea (58%), anosmia (70%), dysgeusia (64%), rhinorrhea (47%), sore throat (52%), cough (77%), dyspnea (56%), nausea and/or vomiting (35%), and abdominal pain (30%). Three of the COVID-19 positive cases were hospitalized; one required ICU care and mechanical ventilation (Fig 1).

We assessed LFA performance using subject serum and compared these to PCR and the immunoassay. CareStart showed 96.2%/89.2% sensitivity/specificity with PCR and 98.3%/94.0% sensitivity/specificity with the immunoassay (Table 2). Healgen showed 95.2%/88.2% sensitivity/specificity with PCR and 96.5%/94.2% sensitivity/specificity with the immunoassay

**Table 1. Participant demographic data.**

|  | All Subjects (n = 512) | Autobio Tested (n = 149) |
|---|---|---|
| **Age (years)** |  |  |
| **Mean (SD)** | 41 (11.6) | 43 (12.3) |
| **Median (IQR)** | 39 (31 to 50) | 43 (32 to 53) |
| **Range** | 19 to 71 | 20 to 71 |
| **Sex (n [%])** |  |  |
| **Female** | 376 (73) | 120 (81) |
| **Male** | 136 (27) | 29 (19) |
| **Race (n [%])** |  |  |
| **Asian** | 50 (10) | 14 (9) |
| **Black or African American** | 22 (4) | 12 (8) |
| **White** | 424 (83) | 117 (79) |
| **More Than One Race** | 13 (3) | 4 (3) |
| **Unknown / Not Reported** | 1 (0) | 1 (1) |
| **Ethnicity (n [%])** |  |  |
| **Hispanic or Latino** | 25 (5) | 9 (6) |
| **Not Hispanic or Latino** | 485 (95) | 138 (93) |
| **Unknown / Not Reported** | 2 (0) | 2 (1) |
| **Michigan Medicine health care worker (n [%])** |  |  |
| **Yes** | 449 (88) | 133 (89) |
| **No** | 63 (12) | 16 (11) |

All subjects' serum was tested using both the CareStart and Healgen assays.

(Table 2). Autobio showed a lower sensitivity/specificity, 87.8%/84.6% with PCR and 93.2%/96.1% with the immunoassay (Table 1) over 149 subjects tested prior to EUA revocation. Notably, among subjects with a negative or no prior PCR, 10 (5%) were positive on CareStart, 11 (6%) were positive on Healgen, and 2 (4%) were positive on Autobio (Fig 2A–2C, Table 2).

These LFA give separate IgG and IgM results. In general, using only IgG results for the LFA produced slightly better sensitivity/specificity results for all three tests, detailed in Table 3. Further, IgM should appear soon after initial infection and decay over time. Among PCR-positive subjects, we plotted time between PCR and LFA using serum against IgM result. For CareStart, there was a trend toward a shorter interval from positive PCR among those with a positive IgM (Fig 2D). For Healgen, there was no difference in time between PCR and LFA testing for IgM status (Fig 2E). For Autobio, IgM positive subjects had a significantly shorter time between PCR and LFA (Fig 2F).

We compared LFA performance using finger-stick blood and serum (Fig 3A–3C). The CareStart showed 96.6%/98.9% sensitivity/specificity between finger-stick whole blood and separated serum and 98.1%/93.4% sensitivity/specificity between finger-stick and the immunoassay (Table 2). The Healgen showed 92.1%/97.0% sensitivity/specificity agreement between finger-stick and serum and 93.2%/94.9% sensitivity/specificity between finger-stick and the immunoassay (Table 2). The Autobio showed 88.4%/97.9% sensitivity/specificity between finger-stick and serum and 90.5%/98.0% sensitivity/specificity between finger-stick and the immunoassay (Table 2). As above, all three LFA showed slightly improved test characteristics using finger-stick blood when IgG was considered alone (Table 3). There was no significant subgroup variation in agreement by age or employment. The CareStart serum vs. POC comparison showed slightly higher agreement among older individuals (AUC = 0.95 among those <40; = 0.99 among those 40+, $p$ = 0.03); all others tests showed no difference by age.

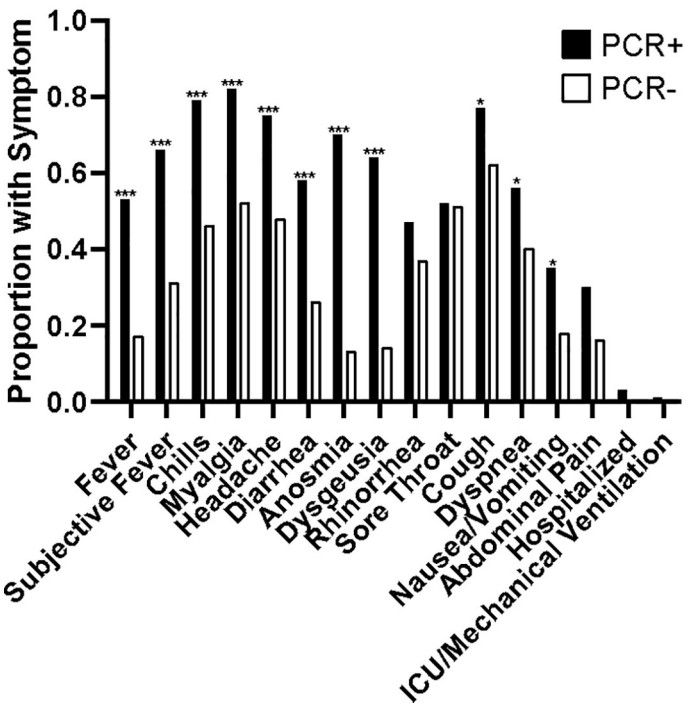

**Fig 1. Symptoms of COVID PCR+ vs PCR- subjects.** Proportion of subjects with symptoms of potential COVID-19 infection among those with a positive PCR and those with a negative PCR. * p < 0.05, *** p < 0.001.

**Table 2. Test comparisons with sensitivity, specificity, and AUC.**

| LFA | Comparator | Sensitivity (TP, FN) | Specificity (FP, TN) | PPV | NPV | AUC (95% CI) |
|---|---|---|---|---|---|---|
| CareStart Serum | PCR | 96.2 (100, 4) | 89.2 (10, 83) | 90.9 | 95.4 | 0.927 (0.890, 0.964) |
| CareStart Serum | Immunoassay | 98.3 (113, 2) | 94.0 (24, 373) | 82.5 | 99.5 | 0.961 (0.944, 0.978) |
| CareStart POC | PCR | 94.9 (93, 5) | 90.6 (5, 48) | 94.9 | 90.6 | 0.927 (0.882, 0.973) |
| CareStart POC | Immunoassay | 98.1 (103, 2) | 93.4 (13, 183) | 88.8 | 98.9 | 0.957 (0.935, 0.979) |
| CareStart POC | CareStart Serum | 96.6 (114, 4) | 98.9 (2, 181) | 98.3 | 97.8 | 0.978 (0.960, 0.996) |
| Healgen Serum | PCR | 95.2 (99, 5) | 88.2 (11, 82) | 90.0 | 94.3 | 0.917 (0.878, 0.956) |
| Healgen Serum | Immunoassay | 96.5 (111, 4) | 94.2 (23, 374) | 82.8 | 98.9 | 0.954 (0.933, 0.974) |
| Healgen POC | PCR | 91.2 (52, 5) | 91.1 (41, 4) | 92.9 | 89.1 | 0.912 (0.856, 0.968) |
| Healgen POC | Immunoassay | 93.2 (55, 4) | 94.9 (7, 130) | 88.7 | 97.0 | 0.941 (0.903, 0.978) |
| Healgen POC | Healgen Serum | 92.1 (58, 5) | 97.0 (4, 129) | 93.5 | 96.3 | 0.945 (0.909, 0.982) |
| Autobio Serum | PCR | 87.8 (36, 5) | 84.6 (2, 11) | 94.7 | 68.8 | 0.862 (0.748, 0.976) |
| Autobio Serum | Immunoassay | 93.2 (41, 3) | 96.1 4, 98) | 91.1 | 97.0 | 0.946 (0.904, 0.988) |
| Autobio POC | PCR | 87.2 (34, 5) | 92.9 (1, 13) | 97.1 | 72.2 | 0.900 (0.812, 0.988) |
| Autobio POC | Immunoassay | 90.5 (38, 4) | 98.0 (2, 98) | 95.0 | 96.1 | 0.942 (0.895, 0.989) |
| Autobio POC | Autobio Serum | 88.4 (38, 5) | 97.9 (94, 2) | 95.0 | 94.9 | 0.931 (0.881, 0.982) |

The LFA column gives the brand of LFA used and the blood component analyzed, either serum or point-of-care (POC) finger-stick-derived blood. The Comparator is either PCR, the immunoassay (using serum) or, where POC finger-stick blood was used, the same LFA using serum was the comparator. Please note, the rows that use PCR as a comparator include only those subjects with a PCR positive or negative result and do not include those with no history of a PCR. TP = true positives, subject number positive on both the test and comparator. FN = false negatives, subject number negative on the test but positive on comparator. FP = false positive, subject number positive on test but negative on comparator. TN = true negative, subject number negative on test and comparator. PPV = Positive Predictive Value.
NPV = Negative Predictive Value.

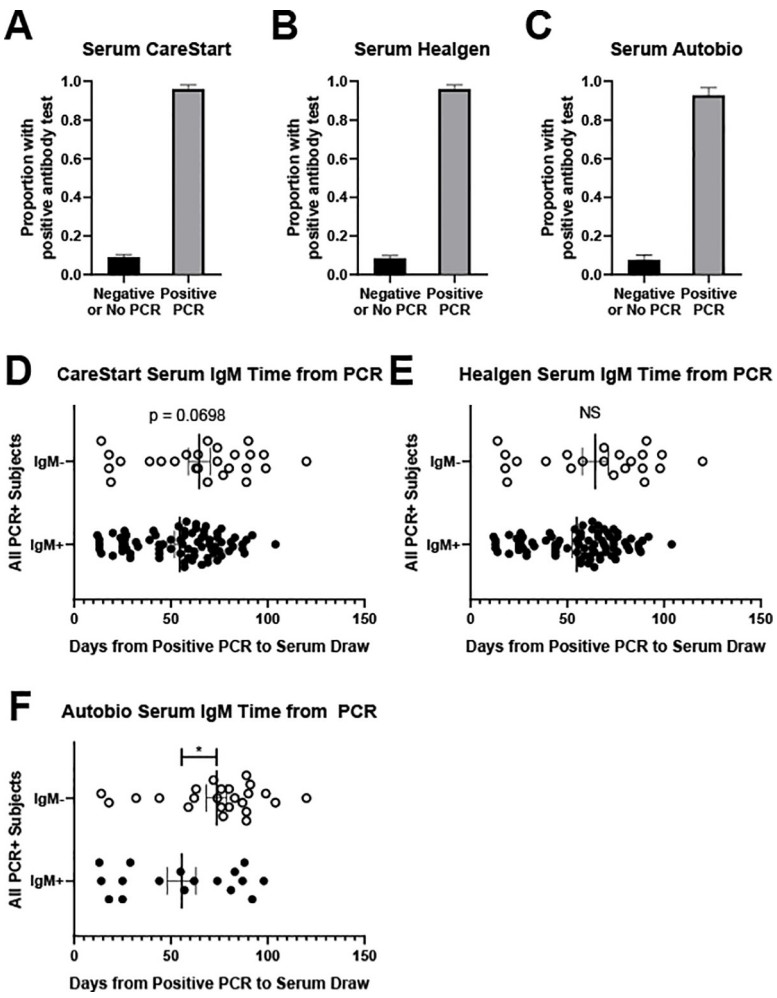

**Fig 2. Results of serum testing for each LFA.** Proportion of known COVID-19 PCR positive subjects and known COVID-19 PCR negative or unknown PCR status with positive LFA result using serum for CareStart (A), Healgen (B) or Autobio (C). IgM status plotted against time from positive PCR to LFA test using serum for CareStart (D), Healgen (E), and Autobio (F). Note, D-F only include subjects with a known PCR-positive result. * p < 0.05. NS = not significant, p = 0.1076.

We also evaluated LFA IgM results among PCR positive subjects using finger-stick. CareStart showed no difference in time between PCR and LFA testing (Fig 3D). Healgen showed significantly more IgM positive subjects when assayed a shorter time between PCR and LFA testing (Fig 3E); the Autobio assay showed a similar but non-significant trend (Fig 3F).

## Discussion

The COVID-19 pandemic has profoundly affected society [11, 12], causing millions of cases and deaths [1]. With the advent of effective vaccines [13], identifying individuals with prior COVID-19 has renewed interest. Importantly, knowing prior COVID-19 infection history could help in vaccine prioritization and evaluating individuals with prior infections and analyzing for vaccine responses and side effects [14, 15]. Despite this, there is a debate whether SARS-CoV-2 antibody can serve as a surrogate for COVID-19 immunity, which remains unresolved in part due to poor testing methods [14, 16]. This was especially true of POC antibody assays, which were initially released without adequate FDA review. While this problem has

**Table 3. Test comparisons using only IgG for LFA.**

| LFA | Comparator | Sensitivity (TP, FN) | Specificity (FP, TN) | PPV | NPV | AUC (95% CI) |
|---|---|---|---|---|---|---|
| CareStart Serum | PCR | 96.2 (100, 4) | 92.5 (7, 86) | 93.5 | 95.6 | 0.943 (0.910, 0.976) |
| CareStart Serum | Immunoassay | 98.3 (113, 2) | 97.2 (11, 386) | 91.1 | 99.5 | 0.977 (0.963, 0.992) |
| CareStart POC | PCR | 93.9 (92, 6) | 92.5 (4, 49) | 95.8 | 89.1 | 0.932 (0.889, 0.975) |
| CareStart POC | Immunoassay | 97.1 (102, 3) | 95.9 (8, 188) | 92.7 | 98.4 | 0.965 (0.944, 0.986) |
| CareStart POC | CareStart Serum | 97.3 (109, 3) | 99.5 (1, 188) | 99.1 | 98.4 | 0.984 (0.968, 0.999) |
| Healgen Serum | PCR | 95.2 (99, 5) | 91.4 (8, 85) | 92.5 | 94.4 | 0.933 (0.898, 0.968) |
| Healgen Serum | Immunoassay | 96.5 (111, 4) | 96.0 (16, 381) | 87.4 | 99.0 | 0.962 (0.943, 0.982) |
| Healgen POC | PCR | 91.2 (52, 5) | 91.1 (4, 41) | 92.9 | 89.1 | 0.912 (0.856, 0.968) |
| Healgen POC | Immunoassay | 93.2 (55, 4) | 94.9 (7, 130) | 88.7 | 97.0 | 0.941 (0.903, 0.978) |
| Healgen POC | Healgen Serum | 93.5 (58, 4) | 97.0 (4, 130) | 93.5 | 97.0 | 0.953 (0.919, 0.987) |
| Autobio Serum | PCR | 87.8 (36, 5) | 84.6 (2, 11) | 94.7 | 68.8 | 0.862 (0.748, 0.976) |
| Autobio Serum | Immunoassay | 93.2 (41, 3) | 97.1 (99, 3) | 93.2 | 97.1 | 0.951 (0.910, 0.992) |
| Autobio POC | PCR | 87.2 (34, 5) | 92.9 (1, 13) | 97.1 | 72.2 | 0.900 (0.812, 0.988) |
| Autobio POC | Immunoassay | 90.5 (38, 4) | 98.0 (2, 98) | 95.0 | 96.1 | 0.942 (0.895, 0.989) |
| Autobio POC | Autobio Serum | 90.5 (38, 4) | 97.9 (95, 2) | 95.0 | 96.0 | 0.942 (0.895, 0.989) |

The LFA column gives the brand of LFA used and the blood component analyzed, either serum or point-of-care (POC) finger-stick-derived blood. The Comparator is either PCR, the immunoassay (using serum), or, where POC finger-stick blood was used, the same LFA using serum was the comparator. Please note, the rows that use PCR as a comparator include only those subjects with a PCR positive or negative result and do not include those with no history of a PCR. TP = true positives, subject number positive on both the test and comparator. FN = false negatives, subject number negative on the test but positive on comparator. FP = false positive, subject number positive on test but negative on comparator. TN = true negative, subject number negative on test and comparator. PPV = Positive Predictive Value. NPV = Negative Predictive Value.

been resolved, the use of antibody as a marker of COVID-19 remains controversial, especially in patients with mild clinical illness.

In this prospective study, we demonstrate that FDA EUA-approved COVID-19 antibody LFA can provide accurate measures of prior COVID-19 infection; this holds true even when the tests are used with finger-stick blood. There are several unique aspects of this study. The population differs from other studies in that most subjects had a mild infection course with few requiring hospitalizations (Fig 1). Indeed, we found that COVID-19 antibody production was an accurate marker of prior infection and noted good concordance between PCR status, IgG identified with the immunoassay, and the LFA using either finger-stick whole blood or serum (Tables 2 and 3). Nearly all subjects with a positive COVID-19 PCR made detectable antibody, which contrasts with studies suggesting less severe infections may not lead to robust antibody responses [17, 18].

The CareStart and Healgen assays showed good concordance using serum-based testing and the immunoassay. There was also good agreement between finger-stick blood and serum analyses for these two assays. This suggests that in this partially unselected population, use of an LFA can detect prior unknown COVID-19 infection to a degree similar to a high-moderate complexity immunoassay. The reasons for the improved performance of antibody screening in this study are likely multiple. We used assays that were fully vetted by the FDA before EUA approval. The Autobio assay we examined that had the EUA revoked had less sensitivity compared to the other tests [9]. In addition, all the tests we employed used spike protein as the primary antigen (Access Bio uses both spike and nucleocapsid proteins) and it has been suggested that nucleocapsid assays have less sensitivity.

This study does have some important limitations. First, nearly 90% of subjects were health-care workers, somewhat limiting the broader applicability of this data. Second, the Autobio

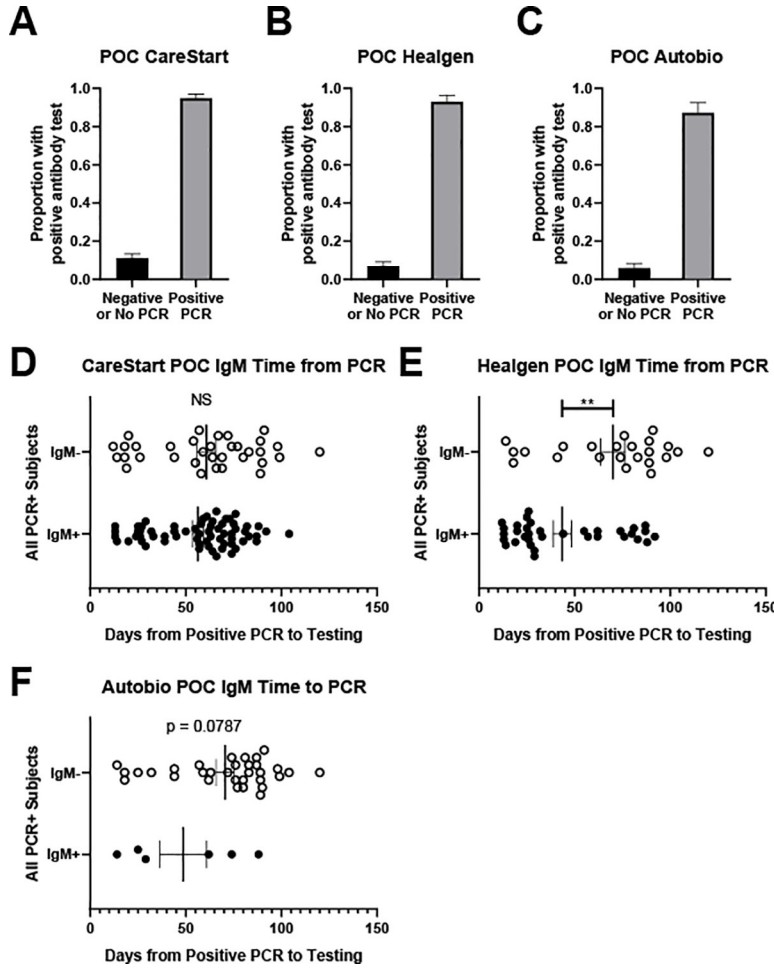

**Fig 3. Results of finger-stick whole blood testing for each LFA.** Proportion of known COVID-19 PCR positive subjects and known COVID-19 PCR negative or unknown PCR status with positive LFA result using finger-stick blood in a point-of-care setting for CareStart (A), Healgen (B) or Autobio (C). IgM status plotted against time from positive PCR to LFA test using finger-stick blood in a point-of-care setting for CareStart (D), Healgen (E), and Autobio (F). Note, D-F only include subjects with a known PCR-positive result. ** p < 0.01. NS = not significant.

sample size was lower than the study sample total due to discontinuing use related to the EUA revocation. Third, one concern with these assays is the false positive rate and specificity, as a false positive could conceivably lead to a false sense of security toward future infection. This could adversely impact risk-taking behavior if the user believes a positive test confirms immunity to future infection, which is not yet known with certainty. One potential source of false positives lies in the concern that pre-existing conditions, such as autoimmune disease or non-SARS-CoV-2 coronaviruses could lead to cross-reactivity and false positives; this is somewhat mitigated by the work required by the FDA for EUA applications and data on the immunoassay from our own institution [8, 10]. Fourth, we did not include immunocompromised subjects, so the applicability of this testing to that population is not known. Finally, cost differences for lab-based testing versus LFA may significantly vary by location and country. In our local area, the clinical cost of a CLIA-lab-based immunoassay is approximately $42 - $119 versus $30 - $50 for an LFA. The LFA cost is expected to decrease in the future.

There remains debate whether SARS-CoV-2 antibodies can serve as a surrogate for COVID-19 immunity, which remains unresolved in part due to poor testing methods [14, 16].

This was especially true of POC antibody assays, which were initially released without minimal FDA review. The test that lost EUA during the course of our study demonstrates that effective corrective action has been taken. These data suggest that regulatory agencies are now appropriately approving these tests but may not indicate that all LFA assays are accurate when using finger- stick blood [19]. Importantly, data are emerging that individuals with antibodies against COVID-19 may be at lower risk of infection [20]. The tests evaluated herein correlate well with higher complexity serum antibody assays, require less infrastructure, and can be performed in POC setting without a venous blood draw. This could allow LFA to provide accurate antibody results in primary care and POC situations and assist in prioritizing individuals receiving COVID-19 vaccines. Such a "screen before vaccine" approach might be helpful in prioritizing vaccines in the general population, where those who produce a positive POC test could conceivably wait for a vaccination. Finally, these assays might also serve to screen for antibody responses to the COVID-19 vaccines, as the antibody concentrations generated after vaccination appear to be substantial [7]; further work would be needed to address this as well.

## Acknowledgments

The authors acknowledge the efforts of Mariam Farida, our study coordinator. The authors acknowledge Bridgette Kaul and Ann Wegmann, both of whom provided significant assistance with subject recruitment.

## Author Contributions

**Conceptualization:** Charles F. Schuler, IV, Carmen Gherasim, Kelly O'Shea, David M. Manthei, Don Giacherio, Jonathan P. Troost, James L. Baldwin, James R. Baker, Jr.

**Data curation:** Charles F. Schuler, IV, Carmen Gherasim, Kelly O'Shea, David M. Manthei, Jesse Chen, Jonathan P. Troost.

**Formal analysis:** Charles F. Schuler, IV, Carmen Gherasim, Kelly O'Shea, David M. Manthei, Jesse Chen, Jonathan P. Troost, James L. Baldwin, James R. Baker, Jr.

**Funding acquisition:** Charles F. Schuler, IV, James L. Baldwin, James R. Baker, Jr.

**Investigation:** Charles F. Schuler, IV, Kelly O'Shea, David M. Manthei, Jesse Chen, James L. Baldwin, James R. Baker, Jr.

**Methodology:** Charles F. Schuler, IV, Carmen Gherasim, Kelly O'Shea, David M. Manthei, Jesse Chen, Jonathan P. Troost, James L. Baldwin, James R. Baker, Jr.

**Project administration:** Charles F. Schuler, IV, Jesse Chen, James L. Baldwin, James R. Baker, Jr.

**Resources:** Carmen Gherasim, David M. Manthei, Jesse Chen, Don Giacherio, Jonathan P. Troost, James L. Baldwin, James R. Baker, Jr.

**Software:** Jonathan P. Troost.

**Supervision:** Don Giacherio, James L. Baldwin, James R. Baker, Jr.

**Validation:** Charles F. Schuler, IV, Carmen Gherasim, Kelly O'Shea, David M. Manthei, Jesse Chen, Jonathan P. Troost.

**Visualization:** Charles F. Schuler, IV, Kelly O'Shea, Don Giacherio, Jonathan P. Troost, James L. Baldwin, James R. Baker, Jr.

**Writing – original draft:** Charles F. Schuler, IV, Jesse Chen, Jonathan P. Troost, James L. Baldwin, James R. Baker, Jr.

**Writing – review & editing:** Charles F. Schuler, IV, Carmen Gherasim, Kelly O'Shea, David M. Manthei, Jesse Chen, Don Giacherio, Jonathan P. Troost, James L. Baldwin, James R. Baker, Jr.

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
