## [Decision Letter · Decision Letter 0]

24 Feb 2021

PONE-D-21-03214

Accurate Point-of-Care Serology Tests for COVID-19

PLOS ONE

Dear Dr. Schuler IV,

Thank you for submitting your manuscript to PLOS ONE. After careful consideration, we feel that it has merit but does not fully meet PLOS ONE’s publication criteria as it currently stands. Therefore, we invite you to submit a revised version of the manuscript that addresses the points raised during the review process.

One of the reviewers questioned whether to include data on the Autobio assay since the assay was terminated half way through the study, however I encourage you to include this information never the less. Kindly include an ethics statement in the manuscript and please revise all references to ensure they match PLoS One requirements (i.e. reference 8 currently only says "FDA")

We look forward to receiving your revised manuscript.

Kind regards,

Benedikt Ley

Academic Editor

PLOS ONE

Journal Requirements:

2)  In your Methods section, please provide additional information about the participant recruitment method and the demographic details of your participants. Please ensure you have provided sufficient details to replicate the analyses such as: a) the recruitment date range (month and year), b) descriptions of where participants were recruited and where the research took place.

3) Please provide a sample size and power calculation in the Methods, or discuss the reasons for not performing one before study initiation.

4) Please ensure you have discussed any potential limitations of your study in the Discussion.

5) To comply with PLOS ONE submission guidelines, in your Methods section, please provide additional information regarding your statistical analyses. For more information on PLOS ONE's expectations for statistical reporting, please see https://journals.plos.org/plosone/s/submission-guidelines.#loc-statistical-reporting.

6)  We note that you have indicated that data from this study are available upon request. PLOS only allows data to be available upon request if there are legal or ethical restrictions on sharing data publicly. For information on unacceptable data access restrictions, please see http://journals.plos.org/plosone/s/data-availability#loc-unacceptable-data-access-restrictions.

7)  Thank you for stating the following in the Acknowledgments Section of your manuscript:

[The authors acknowledge the generosity of Healgen, Access Bio, and Autobio companies who donated the LFA for this study.]

 [The funders had no role in study design, data collection and interpretation, manuscript

drafting, manuscript editing, or the decision to submit the work for publication. This

work was supported by The University of Michigan (Institutional Funding, COVID-19

Innovation Grant, https://www.research.umich.edu/covid-19/covid-19-fundingopportunities),

the National Institutes of Health (UL1TR002240), and through related

sponsored projects from Healgen Scientific (healgen.com) and Access Bio Inc

(accessbiodiagnostics.net).]

Additionally, because some of your funding information pertains to commercial funding, we ask you to provide an updated Competing Interests statement, declaring all sources of commercial funding.

In your Competing Interests statement, please confirm that your commercial funding does not alter your adherence to PLOS ONE Editorial policies and criteria by including the following statement: "This does not alter our adherence to PLOS ONE policies on sharing data and materials.” as detailed online in our guide for authors  http://journals.plos.org/plosone/s/competing-interests.  If this statement is not true and your adherence to PLOS policies on sharing data and materials is altered, please explain how.

Please include the updated Competing Interests Statement and Funding Statement in your cover letter. We will change the online submission form on your behalf.

Reviewers' comments:

Reviewer's Responses to Questions

**Comments to the Author**

1. Is the manuscript technically sound, and do the data support the conclusions?

Reviewer #1: Yes

Reviewer #2: Yes

2. Has the statistical analysis been performed appropriately and rigorously? 

Reviewer #1: Yes

Reviewer #2: Yes

3. Have the authors made all data underlying the findings in their manuscript fully available?

Reviewer #1: No

Reviewer #2: Yes

4. Is the manuscript presented in an intelligible fashion and written in standard English?

Reviewer #1: Yes

Reviewer #2: Yes

5. Review Comments to the Author

Reviewer #1: In general, the manuscript is well written and will be of interest to policy makers and HCPs responding to COVID-19. I’m not entirely convinced the Autobio information should be included since this LFA was discontinued early in the study, but I will leave that up to the editor and submitting authors. Please see below for issues that should be addressed before publishing.

Methods

When did this study occur? Please provide details on enrollment period.

Why did you stop using the Autobio assay? I understand that the EUA approval was revoked but I’m assuming clinical decisions weren’t being made based on POC assays. Why not continue testing your original research question and add to the body of literature about the performance of the Autobio assay?

Results

Please include positive and negative predictive values in results (can also go in supplemental docs)

Line 149 – is it worth including Autobio results as a supplementary? Did the poor performance align with what other studies were finding? Alternatively, should any detail about the Autobio be included in this study given early termination? It might make the entire paper cleaner if all Autobio is removed or sent to appendix .

Did age or occupation (HCW vs general pop) have any impact on POC performance or rates? Please mention.

Discussion

Please describe your interpretation of specificity of these assays. FPs are equally concerning as FNs, possibly even more so, given that a FP might result in a change in behavior, a false sense of security, and an increase in risk taking.

Please add a limitation that you don’t know how the POC will perform in immune compromised individuals.

Can you provide any details or estimates on costs differences between approaches? POC fingerstick will be much cheaper, but capturing the magnitude here will be of value for readers.

Line 263 – is it correct to describe all three LFAs showing good concordance? Above you mention that only CareStart and Healgen were tested against both finger stick and serum. There are also differences in sample sizes, with only 149 tested using Autobio.

You touch on it briefly (line 282), but in my opinion the strongest application of POC testing regarding vaccines would be a “screen before vaccine” approach in areas with limited access to vaccines – where individuals without antibodies are prioritized over those with previous exposure. This is by no means fool proof, but if herd immunity is the ultimate goal, this approach would expedite that process by prioritizing POC – cases, and moving POC +s to the back of the vaccine que. However, such an approach would only be appropriate when vaccinating general population, where high risk/vulnerable populations should receive the vaccine regardless of POC status. When monitoring vaccine adverse events this assay may have some utility, but it should not replace quantitative serological assays that can better describe immune response.

I somewhat disagree with line 284 – LFAs are unable to quantify concentration of antibodies, or if detected antibody confers protective immunity, instead it’s a marker of previous exposure. I think you would need to test these POCs in vaccinated populations before making this statement.

Reviewer #2: Dear Authors,

I am wondering why you'd need to evaluate the LFA that have been FDA approved? However, if your aim is to evaluate these using finger-stick samples for recent/past infection, then this should be highlighted. Not what's written on line 92 of your manuscript.

Furthermore, I have the following comments/questions to your manuscript:

1. Why not use a quantitative test as reference CLIA test rather than a qualitative albeit CLIA immunoassay?

2. In all your results, you combined PCR (-) and no PCR. Since you do get positive response to antibody test, perhaps it's better to differentiate them. No PCR could mean that this person has Covid-19 already, whereas PCR (-) was undetectable viral particle meaning, there should not be any antibody response. This brings it to my next question,

3. How do you ensure that the results you obtained are not due to false positive? The positive result could be due to other coronavirus infection or other factors such as prior pregnancy or autoimmune response?

4. For the no PCR/PCR(-) column, those who were positive for antibody test, I wonder how long has it been since PCR? Since I can't see it from your results.

5. For PCR (+) are all subjects have been infected only once with Sars-CoV-2?

6. Has there been any validation for Roche Elecys anti Sars-CoV-2 against a reference standard? I am just wondering why the authors choose this immunoassay instead of others?

7. I don't understand the difference between Healgen PCR, Healgen serum versus Healgen serum.

8. It's interesting to see if you can have the IgG and IgM put together instead of IgM(-), whether when both are present, they either amplify or subdued the expression of one another.

Thank you.

6. PLOS authors have the option to publish the peer review history of their article (what does this mean?). If published, this will include your full peer review and any attached files.

Reviewer #1: **Yes: **Michael von Fricken

Reviewer #2: No

---

## [Author Response · Author response to Decision Letter 0]

1 Mar 2021

Response to the Reviewers

We thank the reviewers for their hard work in reviewing this manuscript. We have responded to each comment and labeled our responses with “Response:” prior to each response. We hope these changes make the manuscript suitable for publication in PLOS One.

Reviewer #1: In general, the manuscript is well written and will be of interest to policy makers and HCPs responding to COVID-19. I’m not entirely convinced the Autobio information should be included since this LFA was discontinued early in the study, but I will leave that up to the editor and submitting authors. Please see below for issues that should be addressed before publishing.

Response: We thank reviewer one for the thoughtful discussion, particularly regarding Autobio. The academic editor has encouraged us to retain this data, and on discussion within our group, we would like to maintain the data in its current position as a contrast to the other tests as discussed below. 

Methods

When did this study occur? Please provide details on enrollment period.

Response: This study occurred between April and October 2020. We have updated the enrollment information in the Methods as suggested (line 85-86).

Why did you stop using the Autobio assay? I understand that the EUA approval was revoked but I’m assuming clinical decisions weren’t being made based on POC assays. Why not continue testing your original research question and add to the body of literature about the performance of the Autobio assay?

Response: This is an excellent question. Use of the Autobio assay ceased due to loss of EUA approval. The University of Michigan Institutional Review Board, which governed the conduct of this study, mandated a pause in use of the Autobio assay when the EUA was revoked. In order to continue the study, we decided at that time to continue enrollment without the test so as not to delay results otherwise. We added a brief comment on this (line 137). 

Results

Please include positive and negative predictive values in results (can also go in supplemental docs)

Response: We have added positive and negative predictive values to table 2 and table 3 as requested.

Line 149 – is it worth including Autobio results as a supplementary? Did the poor performance align with what other studies were finding? Alternatively, should any detail about the Autobio be included in this study given early termination? It might make the entire paper cleaner if all Autobio is removed or sent to appendix.

Response: This is a fair point and one we considered as well. One of our goals with this manuscript is to address the issues surrounding the large number of point-of-care tests that initially received EUA approval by the US FDA without validation. This led to confusion, mistrust, and difficulty in use of these assays early in the pandemic. We wanted to illustrate that the FDA change to a validated review of these tests (performed independently by the NCI at Frederick, MD) has removed poor performers, such as the Autobio. This means that currently approved lateral flow assays, when used properly, may have useful applications. We have retained this data at the suggestion of the Academic Editor.

Did age or occupation (HCW vs general pop) have any impact on POC performance or rates? Please mention.

Response: Overall, there was a minimal impact of age or occupation, though in the CareStart assay older individuals had slightly higher agreement between the POC and serum testing. We have added this to lines 219-222.

Discussion

Please describe your interpretation of specificity of these assays. FPs are equally concerning as FNs, possibly even more so, given that a FP might result in a change in behavior, a false sense of security, and an increase in risk taking.

Response: We have added an interpretation to the discussion (lines 294-297). We address this very appropriate concern regarding false positive results and resultant behaviors.

Please add a limitation that you don’t know how the POC will perform in immune compromised individuals.

Response: This has been added to lines 301-302. 

Can you provide any details or estimates on costs differences between approaches? POC fingerstick will be much cheaper, but capturing the magnitude here will be of value for readers.

Response: This has been added to lines 302-305.

Line 263 – is it correct to describe all three LFAs showing good concordance? Above you mention that only CareStart and Healgen were tested against both finger stick and serum. There are also differences in sample sizes, with only 149 tested using Autobio.

Response: We have adjusted this in the discussion (lines 281, 287, 292-294) to address this concern. 

You touch on it briefly (line 282), but in my opinion the strongest application of POC testing regarding vaccines would be a “screen before vaccine” approach in areas with limited access to vaccines – where individuals without antibodies are prioritized over those with previous exposure. This is by no means fool proof, but if herd immunity is the ultimate goal, this approach would expedite that process by prioritizing POC – cases, and moving POC +s to the back of the vaccine que. However, such an approach would only be appropriate when vaccinating general population, where high risk/vulnerable populations should receive the vaccine regardless of POC status. When monitoring vaccine adverse events this assay may have some utility, but it should not replace quantitative serological assays that can better describe immune response.

Response: This is an excellent point, and we have added language similar to what you state to the manuscript to enhance this point. Thank you for this suggestion (lines 317-321).

I somewhat disagree with line 284 – LFAs are unable to quantify concentration of antibodies, or if detected antibody confers protective immunity, instead it’s a marker of previous exposure. I think you would need to test these POCs in vaccinated populations before making this statement.

Response: We have clarified the language to suggest further testing is needed in vaccinated populations (now line 319-321).

Reviewer #2: Dear Authors,

I am wondering why you'd need to evaluate the LFA that have been FDA approved? However, if your aim is to evaluate these using finger-stick samples for recent/past infection, then this should be highlighted. Not what's written on line 92 of your manuscript.

Response: The University of Michigan IRB was hesitant to allow us to perform a large study using non-EUA-approved LFA, which is partly why we ultimately used these assays in this study. Also importantly, we wanted to use assays that would actually potentially be used clinically with patients, to give clinicians and researchers an understanding of whether these assays are reliable. We have edited lines 72-73 to clarify this.

Furthermore, I have the following comments/questions to your manuscript:

1. Why not use a quantitative test as reference CLIA test rather than a qualitative albeit CLIA immunoassay?

Response: This is an excellent point. In an ideal setting, we would have used a quantitative test as our reference. This study was conducted from April-October 2020 (line 85 added). While there are differences throughout the world of what’s available, in the USA, there were no quantitative tests at the time the study was done. 

2. In all your results, you combined PCR (-) and no PCR. Since you do get positive response to antibody test, perhaps it's better to differentiate them. No PCR could mean that this person has Covid-19 already, whereas PCR (-) was undetectable viral particle meaning, there should not be any antibody response. This brings it to my next question,

Response: This is a fair question. We should clarify that only Figure 2 A-C and Figure 3 A-C include the “PCR uncertain” subjects, and only the Immunoassay portion of Table 2 and Table 3 include those with no PCR history. We have updated the relevant Figure and Table captions to clarify this point. 

3. How do you ensure that the results you obtained are not due to false positive? The positive result could be due to other coronavirus infection or other factors such as prior pregnancy or autoimmune response?

Response: This concern is appropriate. The EUA applications for the LFA and immunoassay used in this study required the evaluation of pre-pandemic samples, including a subset of HIV+ samples (https://www.fda.gov/medical-devices/coronavirus-disease-2019-covid-19-emergency-use-authorizations-medical-devices/eua-authorized-serology-test-performance), so the concern is somewhat mitigated for the LFA and immunoassay. In addition, our Clinical Pathology Lab has published data on the immunoassay that included pre-pandemic samples (now reference 10). We have added a discussion of this limitation to lines 294-301.

4. For the no PCR/PCR(-) column, those who were positive for antibody test, I wonder how long has it been since PCR? Since I can't see it from your results.

Response: All subjects with a known negative PCR were enrolled within 14 days of their PCR. We have clarified this in line 172.

5. For PCR (+) are all subjects have been infected only once with Sars-CoV-2?

Response: Yes, all subjects were only infected once. We have this to line 173 for clarity.

6. Has there been any validation for Roche Elecys anti Sars-CoV-2 against a reference standard? I am just wondering why the authors choose this immunoassay instead of others?

Response: This assay was chosen because it was available to us through our Clinical Pathology Laboratories when this study started (April 2020) and remained available throughout the study. In addition, our Clinical Pathology Laboratory has published our experience using this as a reference standard, and this reference was added to this manuscript (now reference 10).

7. I don't understand the difference between Healgen PCR, Healgen serum versus Healgen serum.

Response: Table 2 and 3, upon review, were not totally clear in their headings. We have added notations to the table captions to clarify what the comparisons actually are. Thank you for this helpful comment.

8. It's interesting to see if you can have the IgG and IgM put together instead of IgM(-), whether when both are present, they either amplify or subdued the expression of one another.

Response: In Figure 2 D-F and Figure 3 D-F, the IgM+ group also includes IgG. Because there were almost no IgM+ results without IgG+ (only 3 in the CareStart group and none in Healgen/Autobio), unfortunately the suggested comparison is not possible with our dataset. 

We thank both reviewers for their helpful, insightful comments. We hope our responses and changes to the manuscript have improved the manuscript substantially and that it now meets the rigorous publishing requirements for PLOS ONE.

---

## [Editor Report · Decision Letter 1]

4 Mar 2021

Accurate Point-of-Care Serology Tests for COVID-19

PONE-D-21-03214R1

Dear Dr. Schuler IV,

We’re pleased to inform you that your manuscript has been judged scientifically suitable for publication and will be formally accepted for publication once it meets all outstanding technical requirements.

Kind regards,

Benedikt Ley

Academic Editor

PLOS ONE
---

## [Editor Report · Acceptance letter]

8 Mar 2021

PONE-D-21-03214R1 

Accurate point-of-care serology tests for COVID-19 

Dear Dr. Schuler IV:

I'm pleased to inform you that your manuscript has been deemed suitable for publication in PLOS ONE. Congratulations! Your manuscript is now with our production department. 

Kind regards, 

on behalf of

Dr. Benedikt Ley 

Academic Editor

PLOS ONE